# Parent Engagement and Support, Physical Activity, and Academic Performance (PESPAAP): A Proposed Theoretical Model

**DOI:** 10.3390/ijerph16234698

**Published:** 2019-11-26

**Authors:** Ryan D. Burns, Yang Bai, You Fu, Christopher D. Pfledderer, Timothy A. Brusseau

**Affiliations:** 1Department of Health, Kinesiology, and Recreation, University of Utah, Salt Lake City, UT 84112, USA; Yang.Bai@utah.edu (Y.B.); chris.pfledderer@utah.edu (C.D.P.); tim.brusseau@utah.edu (T.A.B.); 2School of Community Health Sciences, University of Nevada Reno, Reno, NV 89557, USA; youf@unr.edu

**Keywords:** achievement, adolescents, children, classroom behavior, cognitive skills, exercise, feedback, mediation

## Abstract

An emerging area of research within public health is the interaction between parents and their children for the promotion of physical activity. Higher levels of daily physical activity may not only improve physical health but also yield better academic performance by improving cognitive skills, classroom behavior, and academic achievement within the pediatric population. However, no theoretical model has yet been proposed to interrelate constructs of parental engagement and support, physical activity, and academic performance within the pediatric population. Here, we: 1) summarize salient research related to pediatric physical activity and academic performance, parents’ physical activity engagement with their children, and the role of parental support in child academic performance; 2) propose a theoretical model interrelating parent physical activity engagement and support, physical activity, and academic performance (PESPAAP); 3) identify features of the proposed model that support its potential merit; and 4) provide potential future research directions and potential analyses that can be undertaken to support, modify, or disprove the proposed theoretical model. The proposed PESPAAP model provides a logically sound model that can be modified or expanded upon to improve applicability and generalizability and can be used as a framework to help align testable hypotheses for studies examining these interrelationships.

## 1. Introduction

Parents play a vital role in their children’s physical, emotional, and mental development [1]. Children who have engaged parents tend to develop better socially and emotionally and tend to perform better at school [2,3,4]. An emerging area of research within public health is the interaction between parents and their children for the promotion of school and out-of-school physical activity [5,6]. It is well known that higher levels of daily physical activity not only improve physical health but may also yield better academic performance by improving cognitive skills, classroom behavior, and academic achievement [7,8,9]. Indeed, several programs have been developed to promote physical activity in children and adolescents in school and out-of-school settings [10]. These programs often recommend that parents and families be engaged with the children to provide them with enjoyable and developmentally appropriate opportunities to participate in physical activity [11,12]. Studies have shown when parents are involved and engaged in the promotion of physical activity for their children, these theory-based programs tend to yield greater efficacy for lifestyle behavior improvement [13,14].

A line of research that manifests from a conceptual extension of these findings is how parents’ active engagement in the promotion of physical activity can positively impact specific facets of their child’s academic performance. Additionally, the link between physical activity and academic performance may be confounded by the level of a parent’s support in their child’s academic pursuits as previous studies have shown parental involvement being a predictor of child academic success [15,16]. Parents’ role in promoting their child’s physical activity behaviors and potentially affecting the relationship between physical activity and academic performance enforces the importance parents have in a child’s health and success at school. Unfortunately, there is no clear conceptual organization of these interrelationships, which makes it difficult to align testable hypotheses to a theoretical framework.

Research in the behavioral sciences is most often guided by theory and hypotheses about observed phenomena. Research not aligning to a theory lacks direction and is said to lack a theoretical framework to base the study upon. A theory is a provisional set of interrelated concepts that structure a systematic view of phenomena for the purpose of expanding or predicting [17,18]. A theory is thought to be provisional because it can never be proven (although it can be supported) but can be disproven through empirical research [17]. Proposed features of a good theory are: uniqueness—being distinguishable from others; conservatism—theory persists until a better theory replaces it; generalizability—applies to many areas; fecundity—a theory that can generate new models and hypotheses; parsimony—relative simplicity; internal consistency—a theory that has identified relationships on the basis of which adequate explanations are rendered; empirical riskiness—refutation of the model must be possible; and abstraction—the theory is independent of time and space, usually achieved by adding more relationships [19].

According to Wacker [19], definition of theory has four components that delineate the scientific term from a possible layman definition of theory. Layman definitions of theory may assume to mean something on paper that does not have empirical evidence. However, according to Wacker [19], scientific theory has four components consisting of variable definitions, domain, relationships, and prediction. In this paper, we aim to address these components of scientific theory by defining variables related to “who and what”, specifying the domain specific conditions in which the theory will hold, specifying reasoning for relationships among variables, and the use of predictive claims of the relationships. Addressing these four components of scientific theory will provide guidelines to answer research questions concerning the relationship of parent physical engagement and support, child and adolescent physical activity, and child and adolescent academic achievement. To the authors’ knowledge, no specific scientific theoretical model has yet been proposed to interrelate constructs of parent engagement and support, physical activity, and academic performance within the pediatric population. Doing so will help provide a theoretical framework for studies addressing research questions within this emerging area of research and provide a base model for future research to expand upon to improve generalizability and applicability. Therefore, the purpose of this manuscript is to develop a theoretical model to interrelate constructs of parent engagement and support, physical activity, and academic performance within the pediatric population. More specifically, we: 1) briefly summarize salient research related to pediatric physical activity and academic performance, parent physical activity engagement with their children, and the role of parental support in child academic performance; 2) propose a theoretical model interrelating parent physical activity engagement and academic support, physical activity, and academic performance; 3) identify features of the proposed model that support its potential merit; and 4) provide future research directions and potential analyses that can be undertaken to support, modify, or disprove the proposed theoretical model. 

## 2. Pediatric Physical Activity and Academic Performance

In recent decades, the positive relationships between health behaviors, particularly physical activity, with cognitive functioning has been well explored within the pediatric population [20,21,22,23]. Both acute and chronic physical activity in addition to the health-related fitness domain of cardiorespiratory endurance correlate with specific domains of academic performance [9]. Physical activity, particularly of higher intensities, may manifest physiological mechanisms at the cellular, molecular, and structural level of the brain to improve cognitive skills [24]. Regular physical activity alters neurogenesis and enhances central nervous system metabolism [25]. Furthermore, habitual and acute physical activity may improve attention span and working memory by altering the neurochemicals serotonin, dopamine, and norepinephrine, along with brain-derived neurotrophic factor, synaptic proteins, and insulin-like growth factor-1 [26,27].

In addition to impacting physiological mechanisms that may improve cognitive skills and abilities [28,29,30], physical activity may also impact behavior in the academic classroom [7,31,32,33]. Indeed, physical activity interventions employed during school hours, such as classroom activity breaks and/or enhanced physical activity/physical education curricula, have shown to yield significant benefits in classroom attention and classroom behavior [34,35]. Potential mechanisms for these positive effects on behavior may be due to moderating psychological arousal yielding an internal psychological state conducive for learning, possibly complementing other physiological, cognitive, emotional, and learning mechanisms [36].

There also may be interplay between physical activity and the health-related fitness domain of cardiorespiratory endurance that links to both cognitive functioning and academic achievement [37,38]. Previous research has found that cardiorespiratory endurance is a mediator between physical activity and specific dimensions of cognitive functioning and academic achievement outcomes [39,40]. However, the relationship between physical activity (a behavior) and cardiorespiratory endurance (a trait) is complex [41], and could be bidirectional in nature [42]. Youth who have higher levels of cardiorespiratory endurance tend to be more physically active throughout the school day [43]. Because academic performance consists of cognitive skills, classroom behavior, and academic achievement [44], students who display higher levels cardiorespiratory endurance tend to have higher levels of physical activity, thus influencing the aforementioned mediators leading to better academic performance [45]. Therefore, the physiological trait of high levels of cardiorespiratory endurance and the behavior of physical activity may both be needed to positively impact academic performance and interventions targeting these two constructs either directly or indirectly may yield efficacy [46]. Indeed, Singh et al. [47] found that across the majority of high-quality randomized control trials, a positive beneficial effect of physical activity interventions on mathematics outcomes was observed in the pediatric population. This finding is supported by an earlier meta-analysis conducted by Alvarez-Bueno et al. [48] finding that physical activity interventions improved academic achievement. These correlational and experimental results do indicate a relationship between higher levels of physical activity and better academic performance in the pediatric population, but how to best increase physical activity behaviors in school and out-of-school settings is a lingering public health issue. Of late, programming utilizing family and specifically parental engagement for the promotion of physical activity has shown promise.

## 3. Parent Physical Activity Engagement

Programs derived for the promotion of pediatric physical activity recommend that parents and families be engaged with children to provide them enjoyable and developmentally appropriate opportunities to participate [11,12]. Correlational studies have shown significant relationships of physical activity and its correlates within parent–child dyads due to similarity of environment, thoughts, and affect [49,50,51,52]. These correlations between parent and child physical activity may be explained by the motivational variables of physical activity enjoyment and self-efficacy [5]. Moderate-to-vigorous physical activity has been found to correlate more strongly during weekdays compared to weekends within obese parent–child dyads [53]. Furthermore, the perceived barriers of physical activity are associated with perceived weight among parent–adolescent dyads [54]. However, because many studies examining dyadic relationships of physical activity are correlational, evidence for directionality and possible bi-directionality within parent-adolescent dyads is often precluded. Results from experimental trials provide the best evidence for family engagement to positively impact child physical activity.

Studies have shown that when parents are involved and engaged in the promotion of physical activity of their children, physical activity interventions tend to yield greater efficacy [13,14]. Meta-analyses of familial dyadic interventions to promote physical activity have shown to yield significant pooled effects [55]. In a recent randomized clinical trial, the 8-week “Dads and Daughters Exercising and Empowered” intervention yielded significant improvements in daughter and father physical activity, daughter motor competence, and decreases in daughter screen time that were characterized by medium-to-large effect sizes (d = 0.4 to 0.8) [56]. Using a meta-analysis of 47 studies, it was found that family-based physical activity are effective for increasing physical activity in children aged 5–12 years [11]. These correlational and experimental findings suggest that parent engagement has potential to improve physical activity and its motivational correlates. However, the extension of this link to impacting academic achievement has yet to be thoroughly examined.

## 4. Role of Parent Academic Support in Child Academic Performance

The link between pediatric physical activity and academic performance has been well supported in the recent literature; however, as recommended by the Centers for Disease Control and Prevention, further testing this relationship is a research priority [45,57]. A potential variable that has yet to be explored as a confounding factor is the parents’ level of academic support. Higher levels of parent academic support may predict a child or adolescent’s academic performance. Parental academic support may also associate with parental physical activity engagement, as these two parental behaviors both reflect positive influences and involvement in a child’s life. This conjecture is partially supported by research reporting findings on the importance parents’ role in predicting a child’s academic achievement. Lara and Saracostti [4] used cluster analysis to derive low, medium, and high level parental academic involvement categories and found that children who have parents with low involvement tend to have low academic achievement. Controlling for socioeconomic status and family size, parental education strongly predicted child academic achievement [58]. Perriel [59] found that children want and expect parent support of their academic pursuits and that parental involvement positively predicts child success at math and reading. Wilder [60] synthesized results of nine meta-analyses examining relationships of parental involvement and child academic achievement and found that the strength of the positive relationships varied depending on student’s achievement assessment type, although the statistical significance of the relationship was consistent across grade levels and ethnic groups. Finally, using a cross-sectional survey and a sample of 2,669 sixth grade students, it was found that a unit increase in parental academic participation associated with a 6%–15% improvement in numeracy scores and a 6%–12% improvement in student literacy scores [61]. These results indicate a link between parental academic support and child academic achievement. However, the correlation between parental academic support and parental physical activity engagement, although logical, has not been thoroughly explored. Incorporating these potential relationships with the link between parental involvement in physical activity promotion may provide a comprehensive framework for understanding these interrelationships.

## 5. Proposed Theory of Parent Engagement and Support, Physical Activity, and Academic Performance (PESPAAP)

### 5.1. Overview

The proposed theoretical model of parent engagement and support, physical activity, and academic performance (PESPAAP) is presented in Figure 1 and the definition of terms of the theoretical model are presented in Table 1. The theoretical model has one primary exogenous construct (parent physical activity engagement), one mediator construct (child and adolescent physical activity), and one outcome (academic performance) that is comprised of cognitive skills and abilities, classroom behavior, and academic achievement [45]. There is a secondary exogenous construct (parent academic support) that correlates with parental physical activity engagement and also affecting child and adolescent academic performance as a potential confounding variable. Finally, there are three feedback loops within the theoretical model—academic performance to parent physical activity engagement, child and adolescent physical activity to parent physical activity engagement, and academic performance to parent academic support. These feedback loops may either positively or negatively affect parent physical activity engagement and/or academic support that may subsequently affect the mediator and/or the outcome within the PESPAAP model. Feedback loops may facilitate positive “spirals of engagement” where higher levels of child and adolescent physical activity and/or academic performance lead to continual physical activity engagement and/or academic support from the parent, which further facilitates positive behavioral and/or academic performance outcomes. Feedback loops could also facilitate a negative “spiral of engagement”, where low levels of physical activity and/or academic performance lead to continual parent physical activity disengagement and/or lower levels of parent academic support, which will subsequently negatively impact physical activity behavior and/or academic performance. The concept of “spirals of engagement” has been proposed in other kinesiology-related conceptual and theoretical frameworks [62]. Specific descriptions of the constructs, pathways, and feedback loops within PESPAAP are provided.

### 5.2. Parent Physical Activity Engagement

Parent physical activity engagement is the lone primary exogenous construct in the proposed theoretical model and serves as an antecedent for child and adolescent physical activity. Parent physical activity engagement is when one or both parents help to improve their child’s opportunities for and participation in physical activity either during school or within out-of-school settings [57]. Parent or child and adolescent perception of parent physical activity engagement can also be part of the construct. Examples of parent physical activity engagement are parent serving as a physical activity opportunity volunteer during school or after school (e.g., recess, physical education, after-school), out-of-school co-play with their child, supporting active transportation by walking or biking to school or after school activities, or being willing and able to transport their child to after school or out-of-school sport or other physical activity programs. Perceptions of parental physical activity engagement, from the parent or the child, may also be of significant relevance. Child and adolescent physical activity and/or academic performance may impact parental physical activity engagement and facilitate either positive or negative spirals of engagement. Parental physical activity engagement may also correlate with other behaviors such as parental academic support, which may in turn affect other relationships within PESPAAP.

### 5.3. Child and Adolescent Physical Activity

Child and adolescent physical activity and correlates is the mediator construct within the proposed PESPAAP theoretical model. Physical activity is any bodily movement that increases caloric expenditure above resting caloric expenditure [63]. Physical activity can be quantified using validated questionnaires, pedometer steps, accelerometer counts, systematic observation methods, or caloric expenditure measured using indirect calorimetry or doubly labeled water [64]. Physical activity can also be quantified according to its intensity, including light, moderate, and vigorous physical activity and the aggregation metric of moderate-to-vigorous physical activity or MVPA. For purposes of the proposed model, to support parsimony and generalizability, the general construct of physical activity could also be represented by motivational correlates such as physical activity enjoyment, self-efficacy, perceived competence, and perceived social support. Observed physical activity and motivational correlates of physical activity may have different relationships with academic performance. Despite that the links between motivational factors such as physical activity enjoyment, self-efficacy, and perceived competence have been established [65,66], although theoretically plausible, the potential link between physical activity motivation and academic performance is not as well explored. Physical activity impacts the outcome of academic performance and may provide a feedback mechanism to parental physical activity engagement.

### 5.4. Academic Performance

Academic performance is comprised of different facets that may influence student success at school, including cognitive skills, classroom behavior, and academic achievement [45]. Cognitive skills consist of executive functioning/control, memory, cognitive flexibility, attitudes, and motivation [67]. Classroom behavior consist of both on-task and off-task behaviors in the academic classroom and can also include student attendance [45,68]. Academic achievement consists of standardized test scoring, grade point average, other formal assessment, and knowledge gained that can be used later in life [45]. Academic performance is directly influenced by child and adolescent physical activity and correlates within the theoretical model and may provide a feedback mechanism to both parental physical activity engagement and/or parental academic support. Academic performance may also be influenced by the level of parental academic support within PESPAAP.

### 5.5. Parent Academic Support

Parent academic support is a potential exogenous construct within the proposed theoretical model that can also relate to child and adolescent academic performance. Parent academic support is when one or both parents share responsibility with teachers to help their children learn and meet educational goals [69]. This positive behavior construct may also associate with parental physical activity engagement. Parent academic support can take many forms including being an academic school volunteer, regularly attending parent–teacher conferences, participating in parent–teacher committees, helping their children with their homework, and setting and communicating academic expectations for their children. Types of parent academic support may also vary depending on a child or adolescent’s age and the academic needs of a respective child. The perception of parent academic support may also be a part of this construct, as the perception of parent academic support is associated with academic performance in adolescents [70,71]. Parent academic support may confound the relationship between child and adolescent physical activity and correlates and academic performance and may also receive positive or negative feedback from the child and adolescent academic performance. In Figure 1, parental academic support’s potential confounding influence is represented by direct relationships between it and child and adolescent physical activity and academic performance.

### 5.6. Feedback Loops

Three feedback loops have been proposed impacting both parental physical activity engagement and parental academic support. These feedback loops may facilitate either positive or negative spirals of engagement that will further facilitate continual high or low levels of child and adolescent physical activity and correlates and/or academic performance. An example of positive feedback facilitating a positive spiral of engagement is a higher level of child or adolescent physical activity (or motivational factors of physical activity) providing positive feedback to parents to continually engage in physical activity behavior with their children. This positive feedback will continually maintain higher levels of physical activity and may support continual higher levels of academic support via the child and adolescent physical activity and academic performance relationship. An example of negative feedback facilitating a negative spiral of engagement is lowered levels of child and adolescent physical activity (or motivational factors of physical activity) providing negative feedback to parents that leads to disengagement in the participation of physical activity with their children. The negative spiral of engagement will lead to continual lower levels of child and adolescent physical activity that may negatively affect academic performance. These positive and negative spirals of engagement can also be applied to academic performance affecting parental physical activity engagement and academic performance affecting parental academic support. Feedback loops and spirals of engagement can be modified or disrupted through intervention targeting salient variables representing a respective construct.

## 6. Features of the Proposed Theoretical Model

Wacker [19] proposed a number of features that identify a good theory: uniqueness, conservatism, generalizability, fecundity, parsimony, internal consistency, empirical riskiness, and abstraction. To the authors knowledge, no theory has yet been proposed to link parent engagement and support, child and adolescent physical activity, and academic performance (*Uniqueness*). Although the proposed theory’s *conservatism* is questionable due to the possibility of modification or refutation through empirical research, it has potential for broad *generalizability* for studies examining these interrelationships by applying to both children and adolescents, families of varying socioeconomic strata, race/ethnicities, and cultures. It is also a model rich with potential research questions that can be addressed using various cross-sectional, longitudinal, experimental, and non-experimental research designs and questions that can be tested using broad moderator variables such as sex, age, and/or socioeconomic status (*Fecundity*). Although mediated mechanisms are present in the proposed theoretical model, we believe *parsimony* does exist within the model by use of a limited number of constructs to explain the interrelationships, partially by collapsing child and adolescent physical activity as a concept that encapsulates both observed physical activity and motivational factors that contribute to its engagement. As described in the prior section, many of the interrelationships have evidence for validity or have logical and plausible explanations of the relationships (*Internal Consistency*); however, there is still much uncertainty regarding the strength and potential impact of the relationships, which manifests a degree of *empirical riskiness*. Finally, *abstraction* is present within the model because there are many smaller and internally consistent concepts/relationships represented and integrated into the larger model (e.g., parent engagement and child physical activity, physical activity and academic performance, parental support and academic performance, potential positive and negative feedback).

## 7. Future Research Directions

A number of future research directions arise from the proposed theoretical model. As stated previously, the link between physical activity and academic performance has been established using cognitive skill, classroom behavior, and academic achievement endpoints [33,46,72]. What is unknown is the strength of the relationship between parent physical activity engagement and academic performance using physical activity (and/or its correlates) as a potential mediator variable. Using parent physical activity engagement as a stimulus and primary antecedent for improving physical activity behaviors that subsequently positively effects specific facets of academic performance is theoretically defensible but has yet to be empirically evaluated using mediation models. Quantifying total, direct, and indirect effects for these relationships should be a priority to support the framework’s potential merit. Additionally, although the use of parental academic support as a correlate of parental physical activity engagement is also theoretically defensible, this also has not been empirically evaluated. Specific to the assessment of physical activity, compositional data analysis (CoDa) has been increasingly used to quantify how proportions of lifestyle and physical activity compositional parts (i.e., sleep, sedentary behaviors, and physical activity intensities) relate to various health-related outcomes. These outcomes have included body composition, cardiorespiratory endurance, cardiometabolic health markers, and gross motor skills [73,74,75,76,77]. However, no research has yet to use CoDa with the aim to examine the associations of specific lifestyle and physical activity compositions with specific facets of academic performance. Furthermore, no research has yet to employ CoDa within the context of a mediated mechanism linking parent physical activity engagement and academic performance.

The feedback mechanisms proposed within the theoretical model have been mostly unexamined, especially the link between academic performance and parent physical activity engagement. Given past research, it is theoretically plausible that higher levels of academic achievement may yield a positive spiral of engagement with parent physical activity promotion for their children [78,79,80]; however, the strength and overall impact of this potential feedback loop has not been thoroughly quantified. It is also theoretically plausible that poor academic performance may lead to parent disengagement for the promotion of physical activity for their child. Many parents are unaware of the beneficial impact physical activity has on academic performance and parent expectations and engagement promote academic achievement in school [81,82]. This disengagement may lead to a negative spiral of parental involvement in physical activity promotion due to the allocation of time use to academic work rather than to physical activity. Indeed, past research has identified that parents perceive academic development as more important for their children than the promotion of physical activity [83]. Other potential research questions could be addressed using the model are to track changes of the parent–child dynamic through primary and secondary schooling, exploring perceptions of parental support as well as direct measures of parental support and exploring differences between perceived and actual parental physical activity engagement and academic support, exploring differences in varying cultural contexts, disruptions in these interrelationships during school holidays and school summer break, and examining differences between mother and father support on the relationship between child and adolescent physical activity and academic performance.

## 8. Potential Future Analyses

The proposed PESPAAP model displays a number of relationships that can be tested in whole or in part. The link between parent physical activity engagement, child and adolescent physical activity, and academic performance can be tested using a mediation model. Within mediation models direct, indirect, and total effects should be tested, including the communication of the percentage of total effect mediated. The link between child and adolescent physical activity and academic performance is potentially confounded by parent academic support. Parent academic support could be used as a control variable if it is indeed thought of as a potential confounder between child and adolescent physical activity and academic performance to hold its effect constant on a respective academic performance outcome. To strengthen and possibly expand the variables used in PESPAAP, use of other mediators may be included in path analyses, including salient physical activity motivational variables in addition to measured physical activity behaviors. Additionally, because child and adolescent physical activity is a broad construct, examining different assessment metrics (e.g., light physical activity, MVPA, total physical activity) and time segments (e.g., weekday, weekend, during school, out-of-school) will provide more specific information, especially within the mediating models. Sedentary behavior may also play a role in these interrelationships. Analyzing both sedentary behaviors and physical activity within the context of a constrained time-use framework can be tested using CoDa. A more comprehensive examination of PESPAAP, including the covariance between parent physical activity engagement and academic support and the latent construct of academic performance, may be addressed using Structural Equation Modeling (SEM) having both structural and measurement components. Bi-directional relationships and feedback mechanisms can be tested using mediation analysis/SEM reversing direction arrows within the model and comparing parameter estimates and model fit accordingly. Longitudinal analysis of these relationships may provide stronger evidence for directionality of effects.

## 9. Conclusions

In this paper we proposed a theoretical model to link parent engagement in the promotion of child and adolescent physical activity with child and adolescent academic performance. We proposed that this link is a potential mediated relationship using observed physical activity behaviors and/or physical activity motivational factors as potential mediators within the relationship pathway. We also proposed that the relationship between child and adolescent physical activity and academic performance may be confounded by parental support for academic performance. Feedback loops were incorporated into the model to identify potential feedback mechanisms facilitating positive or negative spirals of parental engagement. The proposed theoretical model has a number of features that support its potential merit including a theoretical model being rich with potential research questions that can be addressed using a variety of research designs among a variety of different pediatric and adult populations. Our hope is that PESPAAP provides a logically sound model that can be modified or expanded upon and can be used by researchers as a framework to align testable hypotheses involving these interrelationships.

## Figures and Tables

**Figure 1 ijerph-16-04698-f001:**
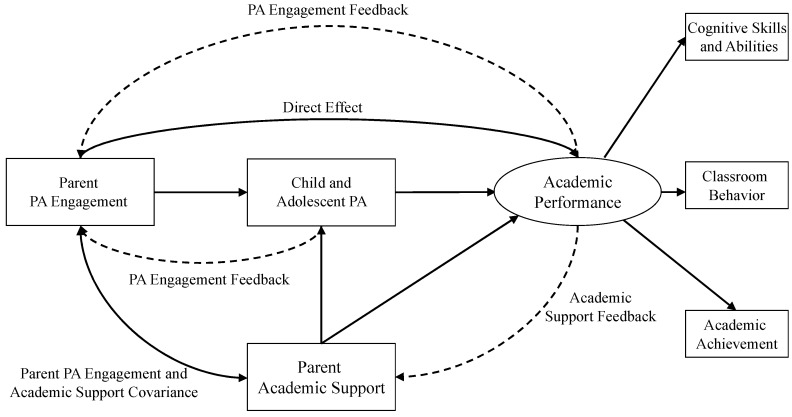
Schematic diagram showing the interrelationships among parent physical activity engagement and academic support, child and adolescent physical activity, and academic performance. *Note:* PA stands for physical activity; dashed lines represent feedback loops.

**Table 1 ijerph-16-04698-t001:** Definition of terms within the proposed theoretical model.

Term	Definition	Examples
Parent Physical Activity Engagement	One or both parents help to increase their child’s opportunities for and participation in physical activity during school and out-of-school.	School physical activity opportunity volunteer; out-of-school co-play; transportation to out-of-school activity or sport programs; active transport encouragement; parental perceptions of physical activity engagement; child and parent perceptions of parental physical activity engagement
Child and Adolescent Physical Activity	Yielding bodily movements that increase energy expenditure above resting energy expenditure. This may also include intra-individual motivational factors that facilitate these movements.	Light physical activity; moderate-to-vigorous physical activity; total physical activity; step counts; activity counts; caloric expenditure; motivational factors (e.g., enjoyment, self-efficacy, perceived competence, perceived social support)
Parent Academic Support	Parents share responsibility with teachers to help their children learn and meet educational goals.	Academic school volunteer; attending parent–teacher conferences; parent–teacher committees; homework participation; expectations; child, adolescent, and parent perceptions of parental academic support
Academic Performance	Different factors that may influence student success at school.	*Cognitive Skills*—executive functioning, attention, memory, attitudes, motivation; *Classroom Behaviors*—positive (on-task) or negative behaviors (off-task) behaviors in the academic classroom, attendance; *Academic Achievement*—standardized test scores, Grade Point Average, other formal assessments; gained knowledge used later in life
Physical Activity Engagement Feedback	Child and adolescent physical activity and academic performance outcomes that have a positive or negative impact on parent physical activity engagement.	Higher levels of child and adolescent physical activity facilitating further parent engagement; negative physical activity outcomes (e.g., lowered total physical activity, lowered enjoyment) decreasing parent engagement. Better academic achievement positively impacting parent physical activity engagement
Academic Support Feedback	Child and adolescent physical academic performance outcomes that have a positive or negative impact on parent academic support.	Better academic performance outcomes may further facilitate parent academic support; poor academic performance may also facilitate increased parent academic support

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
