# Peer review of "Parent Engagement and Support, Physical Activity, and Academic Performance (PESPAAP): A Proposed Theoretical Model"

_ijerph, 2019, doi:10.3390/ijerph16234698_

Round 1

Reviewer 1 Report

This a well-written paper that proposed a conceptual frameworks to interrelate constructs of parent engagement and support, physical activity and academic performance within the pediatric population. It contains a lot of information about the parent engagement in the promotion of child/adolescent physical activity.

1.My major concern is that author shows the interrelationships among physical activity and academic support in theoretical model, yet interrelationship among child/ adolescent PA and parent academic support is lacking. In line 155-157, Author claims that parent academic support may yield stronger magnitude relationships between pediatric physical activity and academic performance, but suitable literature and information on this also is missing.

2.This conceptual frameworks is rather brief and doesn't provide a complete rationale to the interrelationships among variables in the proposed theoretical model.

3.I also have some concerns relating to the variables in this model. Parent PA engagement and parent academic support are independent elements. Parent PA engagement has potential to improve physical activity, Parent academic support associated with improvement in child academic performance, child PA is a potential moderator variable that parent PA engagement impact on the child academic performance, what's the relationship between parent PA engagement and Parent academic support, child PA and parent academic support?

4.Please reformulate some keywords that adequately cover the research and not be appeared in the title.

Author Response

REVIEWER #1

1.My major concern is that author shows the interrelationships among physical activity and academic support in theoretical model, yet interrelationship among child/ adolescent PA and parent academic support is lacking. In line 155-157, Author claims that parent academic support may yield stronger magnitude relationships between pediatric physical activity and academic performance, but suitable literature and information on this also is missing.

-Thank you for this comment. Per this and another Reviewer’s comment, we have decided to eliminate the moderating relationship of parental academic support. In its place, we now conceptualize parental academic support as a potential confounding variable in part because of its relationship with child and adolescent academic achievement. Therefore, we have changed the PESPAAP model (Figure 1) and amended the manuscript text accordingly.

2.This conceptual frameworks is rather brief and doesn't provide a complete rationale to the interrelationships among variables in the proposed theoretical model.

-Thank you. I believe this comment is also related to comment #1 and #3. We have now expanded PESPAAP to include more relationships among the variables/constructs. This includes the potential correlation between parental physical activity engagement and academic support and its potential confounding influence on academic achievement. We elaborate more on these relationships within the manuscript text.

3.I also have some concerns relating to the variables in this model. Parent PA engagement and parent academic support are independent elements. Parent PA engagement has potential to improve physical activity, Parent academic support associated with improvement in child academic performance, child PA is a potential moderator variable that parent PA engagement impact on the child academic performance, what's the relationship between parent PA engagement and Parent academic support, child PA and parent academic support?

-Thank you for this comment. We now add in these potential relationships within a revised PESPAAP model. Specifically, we added the correlation between parental physical activity engagement and academic support and the potential confounding influence of parental academic support, as displayed by direct arrows from it to child and adolescent physical activity and academic performance. We expand on these relationships within the main text of the manuscript.

4.Please reformulate some keywords that adequately cover the research and not be appeared in the title.

-Thank you. New keywords have now been entered in the keyword section.

Reviewer 2 Report

The model presented has the potential to link two important areas: children's physical activity and children's academic performance. The inclusion of parental engagement and support as an intermediary is sound. There are however a few additions which would strengthen the overall manuscript:

The effect of many of these constructs depend not only upon the construct itself but how it is interpreted by other members of the system. For example, parental engagement towards the child's physical activity will also depend on the child's perception of that engagement and how the parent perceives the child's perception. To say nothing about how the parents' perceptions and engagements may impact the behavior of each other. This model may not be as parsimonious as the authors believe.

Judging by the terms used and the model described, structural equation modeling would be how this this model would be analyzed empirically. In that case, a section on probably analyses would strengthen the paper.

lines 50-59 details parental involvement as a moderator. Possibly, this could also be a confounding variable, explaining both academic performance and children's physical activity. If parental involvement is strictly a moderator, explain this role further.

lines 60-84 detail how this model fits into theory. Unfortunately there is the scientific term 'theory' to delineate a well-described set of phenomena and the common laymen term 'theory' which is often assumed to mean something on paper which does not yet have empirical evidence. Ensure that the correct term is used and define how it differs from the layman's term.

Table: academic achievement is not merely about grade earned, but also about gaining knowledge and skills which may be used later in life.

line 86 has an extra word at the beginning of the line

Author Response

REVIEWER #2

The effect of many of these constructs depend not only upon the construct itself but how it is interpreted by other members of the system. For example, parental engagement towards the child's physical activity will also depend on the child's perception of that engagement and how the parent perceives the child's perception. To say nothing about how the parents' perceptions and engagements may impact the behavior of each other. This model may not be as parsimonious as the authors believe.

-Thank you for this helpful comment. Parental perceptions are now part of the parental construct within the PESPAAP model and given as examples within the text and within Table 1. Future research directions examining parental perceptions are now part of the future research directions paragraph within the manuscript.

Judging by the terms used and the model described, structural equation modeling would be how this this model would be analyzed empirically. In that case, a section on probably analyses would strengthen the paper.

-Thank you. A potential data analysis section has now been provided in the manuscript.

lines 50-59 details parental involvement as a moderator. Possibly, this could also be a confounding variable, explaining both academic performance and children's physical activity. If parental involvement is strictly a moderator, explain this role further.

-Thank you for this comment. We have now added in that Parent Academic Support could be a confounding variable with relationship arrows from the variable to child and adolescent PA and academic performance. We now elaborate on this potential relationship within the manuscript. We eliminated parental academic support as a potential moderator because of a lack of sufficient evidence to be included in the model and because of another Reviewer’s comments. Therefore, Figure 1 has changed.

lines 60-84 detail how this model fits into theory. Unfortunately, there is the scientific term 'theory' to delineate a well-described set of phenomena and the common laymen term 'theory' which is often assumed to mean something on paper which does not yet have empirical evidence. Ensure that the correct term is used and define how it differs from the layman's term.

-Thank you for this comment. We now describe four components of scientific theory and why the proposed model fits the scientific definition of the term at the end of the Introduction section.

Table: academic achievement is not merely about grade earned, but also about gaining knowledge and skills which may be used later in life.

-Thank you. This has now been entered into the Table.

line 86 has an extra word at the beginning of the line

-Thank you. This has been revised.

Round 2

Reviewer 1 Report

The revised version is good.